# MaskGAN: Better Text Generation via Filling in the ______

**William Fedus, Ian Goodfellow and Andrew M. Dai**
Google Brain
liam.fedus@gmail.com, {goodfellow, adai}@google.com

## Abstract

Neural text generation models are often autoregressive language models or seq2seq models. These models generate text by sampling words sequentially, with each word conditioned on the previous word, and are state-of-the-art for several machine translation and summarization benchmarks. These benchmarks are often defined by validation perplexity even though this is not a direct measure of the quality of the generated text. Additionally, these models are typically trained via maximum likelihood and teacher forcing. These methods are well-suited to optimizing perplexity but can result in poor sample quality since generating text requires conditioning on sequences of words that may have never been observed at training time. We propose to improve sample quality using Generative Adversarial Networks (GANs), which explicitly train the generator to produce high quality samples and have shown a lot of success in image generation. GANs were originally designed to output differentiable values, so discrete language generation is challenging for them. We claim that validation perplexity alone is not indicative of the quality of text generated by a model. We introduce an actor-critic conditional GAN that fills in missing text conditioned on the surrounding context. We show qualitatively and quantitatively, evidence that this produces more realistic conditional and unconditional text samples compared to a maximum likelihood trained model.

## 1 Introduction

Recurrent Neural Networks (RNNs) (Graves et al., 2012) are the most common generative model for sequences as well as for sequence labeling tasks. They have shown impressive results in language modeling (Mikolov et al., 2010), machine translation (Wu et al., 2016) and text classification (Miyato et al., 2017). Text is typically generated from these models by sampling from a distribution that is conditioned on the previous word and a hidden state that consists of a representation of the words generated so far. These are typically trained with maximum likelihood in an approach known as *teacher forcing*, where ground-truth words are fed back into the model to be conditioned on for generating the following parts of the sentence. This causes problems when, during sample generation, the model is often forced to condition on sequences that were never conditioned on at training time. This leads to unpredictable dynamics in the hidden state of the RNN. Methods such as Professor Forcing (Lamb et al., 2016) and Scheduled Sampling (Bengio et al., 2015) have been proposed to solve this issue. These approaches work indirectly by either causing the hidden state dynamics to become predictable (Professor Forcing) or by randomly conditioning on sampled words at training time, however, they do not directly specify a cost function on the output of the RNN that encourages high sample quality. Our proposed method does so.

Generative Adversarial Networks (GANs) (Goodfellow et al., 2014) are a framework for training generative models in an adversarial setup, with a generator generating images that is trying to fool a discriminator that is trained to discriminate between real and synthetic images. GANs have had a lot of success in producing more realistic images than other approaches but they have only seen limited use for text sequences. This is due to the discrete nature of text making it infeasible to propagate the gradient from the discriminator back to the generator as in standard GAN training. We overcome this by using Reinforcement Learning (RL) to train the generator while the discriminator is still trained via maximum likelihood and stochastic gradient descent. GANs also commonly suffer from issues such

as training instability and mode dropping, both of which are exacerbated in a textual setting. Mode dropping occurs when certain modalities in the training set are rarely generated by the generator, for example, leading all generated images of a volcano to be multiple variants of the same volcano. This becomes a significant problem in text generation since there are many complex modes in the data, ranging from bigrams to short phrases to longer idioms. Training stability is also an issue since unlike image generation, text is generated autoregressively and thus the loss from the discriminator is only observed after a complete sentence has been generated. This problem compounds when generating longer and longer sentences.

We reduce the impact of these problems by training our model on a text fill-in-the-blank or in-filling task. This is similar to the task proposed in Bowman et al. (2016) but we use a more robust setup. In this task, portions of a body of text are deleted or redacted. The goal of the model is to then infill the missing portions of text so that it is indistinguishable from the original data. While in-filling text, the model operates autoregressively over the tokens it has thus far filled in, as in standard language modeling, while conditioning on the true known context. If the entire body of text is redacted, then this reduces to language modeling.

Designing error attribution per time step has been noted to be important in prior natural language GAN research (Yu et al., 2017; Li et al., 2017). The text infilling task naturally achieves this consideration since our discriminator will evaluate each token and thus provide a fine-grained supervision signal to the generator. Consider, for instance, if the generator produces a sequence perfectly matching the data distribution over the first $t - 1$ time-steps, but then produces an outlier token $y_t$, $(x_{1:t-1}y_t)$. Despite the entire sequence now being clearly synthetic as a result of the errant token, a discriminative model that produces a high loss signal to the outlier token, but not to the others, will likely yield a more informative error signal to the generator.

This research also opens further inquiry of conditional GAN models in the context of natural language.

In the following sections,

- We introduce a text generation model trained on in-filling (MaskGAN).

- Consider the actor-critic architecture in extremely large action spaces.

- Consider new evaluation metrics and the generation of synthetic training data.

## 2 RELATED WORK

Research into reliably extending GAN training to discrete spaces and discrete sequences has been a highly active area. GAN training in a continuous setting allows for fully differentiable computations, permitting gradients to be passed through the discriminator to the generator. Discrete elements break this differentiability, leading researchers to either avoid the issue and reformulate the problem, work in the continuous domain or to consider RL methods.

SeqGAN (Yu et al., 2017) trains a language model by using policy gradients to train the generator to fool a CNN-based discriminator that discriminates between real and synthetic text. Both the generator and discriminator are pretrained on real and fake data before the phase of training with policy gradients. During training they then do Monte Carlo rollouts in order to get a useful loss signal per word. Follow-up work then demonstrated text generation without pretraining with RNNs (Press et al., 2017). Additionally (Zhang et al., 2017) produced results with an RNN generator by matching high- dimensional latent representations.

Professor Forcing (Lamb et al., 2016) is an alternative to training an RNN with teacher forcing by using a discriminator to discriminate the hidden states of a generator RNN that is conditioned on real and synthetic samples. Since the discriminator only operates on hidden states, gradients can be passed through to the generator so that the hidden state dynamics at inference time follow those at training time.

GANs have been applied to dialogue generation (Li et al., 2017) showing improvements in adversarial evaluation and good results with human evaluation compared to a maximum likelihood trained baseline. Their method applies REINFORCE with Monte Carlo sampling on the generator.

Replacing the non-differentiable sampling operations with efficient gradient approximators (Jang et al., 2017)has not yet shown strong results with discrete GANs. Recent unbiased and low variance gradient estimate techniques such as Tucker et al. (2017) may prove more effective.

WGAN-GP (Gulrajani et al., 2017) avoids the issue of dealing with backpropagating through discrete nodes by generating text in a one-shot manner using a 1D convolutional network. Hjelm et al. (2017) proposes an algorithmic solution and uses a boundary-seeking GAN objective along with importance sampling to generate text. In Rajeswar et al. (2017), the discriminator operates directly on the continuous probabilistic output of the generator. However, to accomplish this, they recast the traditional autoregressive sampling of the text since the inputs to the RNN are predetermined. Che et al. (2017) instead optimize a lower-variance objective using the discriminator's output, rather than the standard GAN objective.

Reinforcement learning methods have been explored successfully in natural language. Using a REINFORCE and cross entropy hybrid, MIXER, (Ranzato et al., 2015) directly optimized BLEU score and demonstrated improvements over baselines. More recently, actor-critic methods in natural language were explored in Bahdanau et al. (2017) where instead of having rewards supplied by a discriminator in an adversarial setting, the rewards are task-specific scores such as BLEU.

Conditional text generation via GAN training has been explored in Rajeswar et al. (2017); Li et al. (2017).

Our work is distinct in that we employ an actor-critic training procedure on a task designed to provide rewards at every time step (Li et al., 2017). We believe the in-filling may mitigate the problem of severe mode-collapse. This task is also harder for the discriminator which reduces the risk of the generator contending with a near-perfect discriminator. The critic in our method helps the generator converge more rapidly by reducing the high-variance of the gradient updates in an extremely high action-space environment when operating at word-level in natural language.

## 3 MASKGAN

### 3.1 NOTATION

Let $(x_t, y_t)$ denote pairs of input and target tokens. Let <m> denote a masked token (where the original token is replaced with a hidden token) and let $\hat{x}_t$ denote the filled-in token. Finally, $\tilde{x}_t$ is a filled-in token passed to the discriminator which may be either real or fake.

### 3.2 ARCHITECTURE

The task of imputing missing tokens requires that our MaskGAN architecture condition on information from both the past and the future. We choose to use a seq2seq (Sutskever et al., 2014) architecture. Our generator consists of an encoding module and decoding module. For a discrete sequence $\boldsymbol{x} = (x_1, \cdots, x_T)$, a binary mask is generated (deterministically or stochastically) of the same length $\boldsymbol{m} = (m_1, \cdots, m_T)$ where each $m_t \in \{0, 1\}$, selects which tokens will remain. The token at time $t$, $x_t$ is then replaced with a special mask token <m> if the mask is 0 and remains unchanged if the mask is 1.

The encoder reads in the masked sequence, which we denote as $\boldsymbol{m}(\boldsymbol{x})$, where the mask is applied element-wise. The encoder provides access to future context for the MaskGAN during decoding.

As in standard language-modeling, the decoder fills in the missing tokens auto-regressively, however, it is now conditioned on both the masked text $\boldsymbol{m}(\boldsymbol{x})$ as well as what it has filled-in up to that point. The generator decomposes the distribution over the sequence into an ordered conditional sequence $P(\hat{x}_1, \cdots, \hat{x}_T | \boldsymbol{m}(\boldsymbol{x})) = \prod_{t=1}^{T} P(\hat{x}_t | \hat{x}_1, \cdots, \hat{x}_{t-1}, \boldsymbol{m}(\boldsymbol{x}))$.

$$G(x_t) \equiv P(\hat{x}_t | \hat{x}_1, \cdots, \hat{x}_{t-1}, \boldsymbol{m}(\boldsymbol{x})) \tag{1}$$

The discriminator has an identical architecture to the generator[1] except that the output is a scalar probability at each time point, rather than a distribution over the vocabulary size. The discriminator is given the filled-in sequence from the generator, but importantly, it is given the original

---

[1]We also tried CNN-based discriminators but found that LSTMs performed the best.

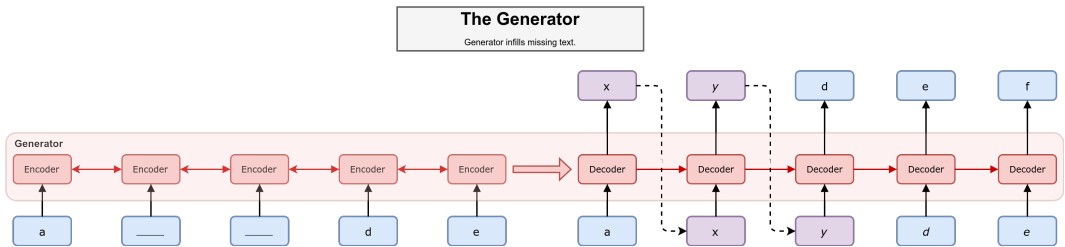

Figure 1: seq2seq generator architecture. Blue boxes represent known tokens and purple boxes are imputed tokens. We demonstrate a sampling operation via the dotted line. The encoder reads in a masked sequence, where masked tokens are denoted by an underscore, and then the decoder imputes the missing tokens by using the encoder hidden states. In this example, the generator should fill in the alphabetical ordering, (a,b,c,d,e).

real context $\boldsymbol{m}(\boldsymbol{x})$. We give the discriminator the true context, otherwise, this algorithm has a critical failure mode. For instance, without this context, if the discriminator is given the filled-in sequence `the director director guided the series`, it will fail to reliably identify the `director director` bigram as fake text, despite this bigram potentially never appearing in the training corpus (aside from an errant typo). The reason is that it is ambiguous which of the two occurrences of `director` is fake; `the *associate* director guided the series` or `the director *expertly* guided the series` are both potentially valid sequences. Without the context of which words are real, the discriminator was found to assign equal probability to both words. The result, of course, is an inaccurate learning signal to the generator which will not be correctly penalized for producing these bigrams. To prevent this, our discriminator $D_\phi$ computes the probability of each token $\tilde{x}_t$ being real given the *true context* of the masked sequence $\boldsymbol{m}(\boldsymbol{x})$.

$$D_\phi(\tilde{x}_t|\tilde{x}_{0:T}, \boldsymbol{m}(\boldsymbol{x})) = P(\tilde{x}_t = x_t^{\text{real}}|\tilde{x}_{0:T}, \boldsymbol{m}(\boldsymbol{x})) \qquad (2)$$

In our formulation, the logarithm of the discriminator estimates are regarded as the reward

$$r_t \equiv \log D_\phi(\tilde{x}_t|\tilde{x}_{0:T}, \boldsymbol{m}(\boldsymbol{x})) \qquad (3)$$

Our third network is the critic network, which is implemented as an additional head off the discriminator. The critic estimates the value function, which is the discounted total return of the filled-in sequence $R_t = \sum_{s=t}^{T} \gamma^s r_s$, where $\gamma$ is the discount factor at each position in the sequence.[2]

## 3.3 TRAINING

Our model is not fully-differentiable due to the sampling operations on the generator's probability distribution to produce the next token. Therefore, to train the generator, we estimate the gradient with respect to its parameters $\theta$ via policy gradients (Sutton et al., 2000). Reinforcement learning was first employed to GANs for language modeling in Yu et al. (2017). Analogously, here the generator seeks to maximize the cumulative total reward $R = \sum_{t=1}^{T} R_t$. We optimize the parameters of the generator, $\theta$, by performing gradient ascent on $\mathbb{E}_{G(\theta)}[R]$. Using one of the REINFORCE family of algorithms, we can find an unbiased estimator of this as $\nabla_\theta \mathbb{E}_G[R_t] = R_t \nabla_\theta \log G_\theta(\hat{x}_t)$. The variance of this gradient estimator may be reduced by using the learned value function as a baseline $b_t = V^G(x_{1:t})$ which is produced by the critic. This results in the generator gradient contribution for a *single* token $\hat{x}_t$

$$\nabla_\theta \mathbb{E}_G[R_t] = (R_t - b_t)\nabla_\theta \log G_\theta(\hat{x}_t) \qquad (4)$$

In the nomenclature of RL, the quantity $(R_t - b_t)$ may be interpreted as an estimate of the advantage $A(a_t, s_t) = Q(a_t, s_t) - V(s_t)$. Here, the action $a_t$ is the token chosen by the generator $a_t \equiv \hat{x}_t$ and

---

[2]MaskGAN source code available at: `https://github.com/tensorflow/models/tree/master/research/maskgan`

the state $s_t$ are the current tokens produced up to that point $s_t \equiv \hat{x}_1, \cdots, \hat{x}_{t-1}$. This approach is an actor-critic architecture where $G$ determines the policy $\pi(s_t)$ and the baseline $b_t$ is the critic (Sutton & Barto, 1998; Degris et al., 2012).

For this task, we design rewards at each time step for a single sequence in order to aid with credit assignment (Li et al., 2017). As a result, a token generated at time-step $t$ will influence the rewards received at that time step and subsequent time steps. Our gradient for the generator will include contributions for each token filled in order to maximize the discounted total return $R = \sum_{t=1}^{T} R_t$.

The full generator gradient is given by Equation 6

$$\nabla_\theta \mathbb{E}[R] = \mathbb{E}_{\hat{x}_t \sim G} \left[ \sum_{t=1}^{T} (R_t - b_t) \nabla_\theta \log(G_\theta(\hat{x}_t)) \right] \tag{5}$$

$$= \mathbb{E}_{\hat{x}_t \sim G} \left[ \sum_{t=1}^{T} \left( \sum_{s=t}^{T} \gamma^s r_s - b_t \right) \nabla_\theta \log(G_\theta(\hat{x}_t)) \right] \tag{6}$$

Intuitively this shows that the gradient to the generator associated with producing $\hat{x}_t$ will depend on *all* the discounted future rewards ($s \geq t$) assigned by the discriminator. For a non-zero $\lambda$ discount factor, the generator will be penalized for greedily selecting a token that earns a high reward at that time-step alone. Then for one full sequence, we sum over all generated words $\hat{x}_t$ for $t = 1 : T$.

Finally, as in conventional GAN training, our discriminator will be updated according to the gradient

$$\nabla_\phi \frac{1}{m} \sum_{i=1}^{m} \left[ \log D(x^{(i)}) \right] + \log(1 - D(G(z^{(i)})) \right] \tag{7}$$

## 3.4 Alternative Approaches for Long Sequences and Large Vocabularies

As an aside for other avenues we explored, we highlight two particular problems of this task and plausible remedies. This task becomes more difficult with long sequences and with large vocabularies. To address the issue of extended sequence length, we modify the core algorithm with a dynamic task. We apply our algorithm up to a maximum sequence length $T$, however, upon satisfying a convergence criterion, we then *increment* the maximum sequence length to $T + 1$ and continue training. This allows the model to build up an ability to capture dependencies over shorter sequences before moving to longer dependencies as a form of curriculum learning.

In order to alleviate issues of variance with REINFORCE methods in a large vocabulary size, we consider a simple modification. At each time-step, instead of generating a reward only on the sampled token, we instead seek to use the full information of the generator distribution. Before sampling, the generator produces a probability distribution over all tokens $G(v) \; \forall \; v \in \mathcal{V}$. We compute the reward for each *possible* token $v$, conditioned on what had been generated before. This incurs a computational penalty since the discriminator must now be used to predict over all tokens, but if performed efficiently, the potential reduction in variance could be beneficial.

## 3.5 Method Details

Prior to training, we first perform pretraining. First we train a language model using standard maximum likelihood training. We then use the pretrained language model weights for the seq2seq encoder and decoder modules. With these language models, we now pretrain the seq2seq model on the in-filling task using maximum likelihood, in particular, the attention parameters as described in Luong et al. (2015). We select the model producing the lowest validation perplexity on the masked task via a hyperparameter sweep over 500 runs. Initial algorithms did not include a critic, but we found that the inclusion of the critic decreased the variance of our gradient estimates by an order of magnitude which substantially improved training.

## 4    EVALUATION

Evaluation of generative models continues to be an open-ended research question. We seek heuristic metrics that we believe will be correlated with human-evaluation. BLEU score (Papineni et al., 2002) is used extensively in machine translation where one can compare the quality of candidate translations from the reference. Motivated by this metric, we compute the number of unique $n$-grams produced by the generator that occur in the validation corpus for small $n$. Then we compute the geometric average over these metrics to get a unified view of the performance of the generator.

From our maximum-likelihood trained benchmark, we were able to find GAN hyperparameter configurations that led to small decreases in validation perplexity on $O(1)-$point. However, we found that these models did not yield considerable improvements to the sample quality so we abandoned trying to reduce validation perplexity. One of the biggest advantages of GAN-trained NLP models, is that the generator can produce alternative, yet realistic language samples, but not be unfairly penalized by not producing with high likelihood the single correct sequence. As the generator explores 'off-manifold' in the free-running mode, it may find alternative options that are valid, but do not maximize the probability of the underlying sequence. We therefore choose not to focus on architectures or hyperparameter configurations that led to small reductions in validation perplexity, but rather, searched for those that improved our heuristic evaluation metrics.

## 5    EXPERIMENTS

We present both conditional and unconditional samples generated on the PTB and IMDB data sets at word-level. MaskGAN refers to our GAN-trained variant and MaskMLE refers to our maximum-likelihood trained variant. Additional samples are supplied in Appendix B.

### 5.1    THE PENN TREEBANK (PTB)

The Penn Treebank dataset (Marcus et al., 1993) has a vocabulary of 10,000 unique words. The training set contains 930,000 words, the validation set contains 74,000 words and the test set contains 82,000 words. For our experiments, we train on the training partition.

We first pretrain the commonly-used variational LSTM language model with parameter dimensions common to MaskGAN following Gal & Ghahramani (2016) to a validation perplexity of 78. After then loading the weights from the language model into the MaskGAN generator we further pretrain with a masking rate of $0.5$ (half the text blanked) to a validation perplexity of 55.3. Finally, we then pretrain the discriminator on the samples produced from the current generator and real training text.

#### 5.1.1    CONDITIONAL SAMPLES

We produce samples conditioned on surrounding text in Table 1. Underlined sections of text are missing and have been filled in via either the MaskGAN or MaskMLE algorithm.

| Ground Truth | **the next day 's show \<eos\> interactive telephone technology has taken a new leap in \<unk\> and television programmers are** |
|---|---|
| MaskGAN | the next day 's show \<eos\> interactive telephone technology has taken a new leap in its retail business \<eos\> a |
| MaskMLE | the next day 's show \<eos\> interactive telephone technology has taken a new leap in the complicate case of the |

Table 1: Conditional samples from PTB for both MaskGAN and MaskMLE models.

### 5.1.2 LANGUAGE MODEL (UNCONDITIONAL) SAMPLES

We may also run MaskGAN in an unconditional mode, where the entire context is blanked out, thus making it equivalent to a language model. We present a length-20 language model sample in Table 2 and additional samples are included in the Appendix.

| | |
|---|---|
| MaskGAN | oct. N as the end of the year the resignations were approved <eos> the march N N <unk> was down |

Table 2: Language model (unconditional) sample from PTB for MaskGAN.

## 5.2 IMDB MOVIE DATASET

The IMDB dataset Maas et al. (2011) consists of 100,000 movie reviews taken from IMDB. Each review may contain several sentences. The dataset is divided into 25,000 labeled training instances, 25,000 labeled test instances and 50,000 unlabeled training instances. The label indicates the sentiment of the review and may be either positive or negative. We use the first 40 words of each review in the training set to train our models, which leads to a dataset of 3 million words.

Identical to the training process in PTB, we pretrain a language model to a validation perplexity of 105.6. After then loading the weights from the language model into the MaskGAN generator we further pretrain with masking rate of $0.5$ (half the text blanked) to a validation perplexity of 87.1. Finally, we then pretrain the discriminator on the samples produced from the current generator and real training text.

### 5.2.1 CONDITIONAL SAMPLES

Here we compare MaskGAN and MaskMLE conditional language generation ability for the IMDB dataset.

| | |
|---|---|
| **Ground Truth** | **Pitch Black was a complete shock to me when I first saw it back in 2000 In the previous years I** |
| MaskGAN | Pitch Black was a complete shock to me when I first saw it back in 1979 I was really looking forward |
| MaskMLE | Black was a complete shock to me when I first saw it back in 1969 I live in New Zealand |

Table 3: Conditional samples from IMDB for both MaskGAN and MaskMLE models.

### 5.2.2 LANGUAGE MODEL (UNCONDITIONAL) SAMPLES

As in the case with PTB, we generate IMDB samples unconditionally, equivalent to a language model. We present a length-40 sample in Table 4 and additional samples are included in the Appendix.

| | |
|---|---|
| MaskGAN | **Positive**: Follow the Good Earth movie linked Vacation is a comedy that credited against the modern day era yarns which has helpful something to the modern day s best It is an interesting drama based on a story of the famed |

Table 4: Language model (unconditional) sample from IMDB for MaskGAN.

## 5.3 PERPLEXITY OF GENERATED SAMPLES

As of this date, GAN training has not achieved state-of-the-art word level validation perplexity on the Penn Treebank dataset. Rather, the top performing models are still maximum-likelihood trained

| Model | Perplexity of IMDB samples under a pretrained LM |
|---|---|
| MaskMLE | $273.1 \pm 3.5$ |
| MaskGAN | $108.3 \pm 3.5$ |

Table 5: The perplexity is calculated using a pre-trained language model that is equivalent to the decoder (in terms of architecture and size) used in the MaskMLE and MaskGAN models. This language model was used to initialize both models.

models, such as the recent architectures found via neural architecture search in Zoph & Le (2017). An extensive hyperparameter search with MaskGAN further supported that GAN training does not improve the validation perplexity results set via state-of-the-art models. However, we instead seek to understand the quality of the *sample generation*. As highlighted earlier, a fundamental problem of generating in free-running mode potentially leads to 'off-manifold' sequences which can result in poor sample quality for teacher-forced models. We seek to quantitatively evaluate this dynamic present only during sampling. This is commonly done with BLEU but as shown by Wu et al. (2016), BLEU is not necessarily correlated with sample quality. We believe the correlation may be even less in the in-filling task since there are many potential valid in-fillings and BLEU would penalize valid ones.

Instead, we calculate the perplexity of the generated samples by MaskGAN and MaskMLE by using the language model that was used to initialize MaskGAN and MaskMLE. Both MaskGAN and MaskMLE produce samples autoregressively (free-running mode), building upon the previously sampled tokens to produce the distribution over the next.

The MaskGAN model produces samples which are more likely under the initial model than the MaskMLE model. The MaskMLE model generates improbable sentences, as assessed by the initial language model, during inference as compounding sampling errors result in a recurrent hidden states that are never seen during teacher forcing (Lamb et al., 2016). Conversely, the MaskGAN model operates in a free-running mode while training and this supports that it is more robust to these sampling perturbations.

## 5.4 MODE COLLAPSE

In contrast to image generation, mode collapse can be measured by directly calculating certain n-gram statistics. In this instance, we measure mode collapse by the percentage of unique n-grams in a set of 10,000 generated IMDB movie reviews. We unconditionally generate each sample (consisting of 40 words). This results in almost 400K total bi/tri/quad-grams.

| Model | % Unique bigrams | % Unique trigrams | % Unique quadgrams |
|---|---|---|---|
| LM | 40.6 | 75.2 | 91.9 |
| MaskMLE | 43.6 | 77.4 | 92.6 |
| MaskGAN | 38.2 | 70.7 | 88.2 |

Table 6: Diversity statistics within 1000 unconditional samples of PTB news snippets (20 words each).

The results in Table 6 show that MaskGAN does show some mode collapse, evidenced by the reduced number of unique quadgrams. However, all complete samples (taken as a sequence) for all the models are still unique. We also observed during RL training an initial small drop in perplexity on the ground-truth validation set but then a steady increase in perplexity as training progressed. Despite this, sample quality remained relatively consistent. The final samples were generated from a model that had a perplexity on the ground-truth of 400. We hypothesize that mode dropping is occurring near the tail end of sequences since generated samples are unlikely to generate all the previous words correctly in order to properly model the distribution over words at the tail. Theis et al. (2016) also shows how validation perplexity does not necessarily correlate with sample quality.

## 5.5 HUMAN EVALUATION

Ultimately, the evaluation of generative models is still best measured by unbiased human evaluation. Therefore, we evaluate the quality of the generated samples of our initial language model (LM), the MaskMLE model and the MaskGAN model in a blind heads-up comparison using Amazon Mechanical Turk. Note that these models have the same number of parameters at inference time. We pay raters to compare the quality of two extracts along 3 axes (grammaticality, topicality and overall quality). They are asked if the first extract, second extract or neither is higher quality.

| Preferred Model | Grammaticality % | Topicality % | Overall % |
| --- | --- | --- | --- |
| LM | 15.3 | 19.7 | 15.7 |
| **MaskGAN** | 59.7 | 58.3 | 58.0 |
| LM | 20.0 | 28.3 | 21.7 |
| **MaskMLE** | 42.7 | 43.7 | 40.3 |
| **MaskGAN** | 49.7 | 43.7 | 44.3 |
| MaskMLE | 18.7 | 20.3 | 18.3 |
| Real samples | 78.3 | 72.0 | 73.3 |
| LM | 6.7 | 7.0 | 6.3 |
| Real samples | 65.7 | 59.3 | 62.3 |
| MaskGAN | 18.0 | 20.0 | 16.7 |

Table 7: A Mechanical Turk blind heads-up evaluation between pairs of models trained on IMDB reviews. 100 reviews (each 40 words long) from each model are unconditionally sampled and randomized. Raters are asked which sample is preferred between each pair. 300 ratings were obtained for each model pair comparison.

| Preferred model | Grammaticality % | Topicality % | Overall % |
| --- | --- | --- | --- |
| LM | 32.0 | 30.7 | 27.3 |
| **MaskGAN** | 41.0 | 39.0 | 35.3 |
| **LM** | 32.7 | 34.7 | 32.0 |
| MaskMLE | 37.3 | 33.3 | 31.3 |
| **MaskGAN** | 44.7 | 33.3 | 35.0 |
| MaskMLE | 28.0 | 28.3 | 26.3 |
| **SeqGAN** | 38.7 | 34.0 | 30.7 |
| MaskMLE | 33.3 | 28.3 | 27.3 |
| SeqGAN | 31.7 | 34.7 | 32.0 |
| **MaskGAN** | 43.3 | 37.3 | 37.0 |

Table 8: A Mechanical Turk blind heads-up evaluation between pairs of models trained on PTB. 100 news snippets (each 20 words long) from each model are unconditionally sampled and randomized. Raters are asked which sample is preferred between each pair. 300 ratings were obtained for each model pair comparison.

The Mechanical Turk results show that MaskGAN generates superior human-looking samples to MaskMLE on the IMDB dataset. However, on the smaller PTB dataset (with 20 word instead of 40 word samples), the results are closer. We also show results with SeqGAN (trained with the same network size and vocabulary size) as MaskGAN, which show that MaskGAN produces superior samples to SeqGAN.

# 6 DISCUSSION

Our work further supports the case for matching the training and inference procedures in order to produce higher quality language samples. The MaskGAN algorithm directly achieves this through GAN-training and improved the generated samples as assessed by human evaluators.

In our experiments, we generally found training where contiguous blocks of words were masked produced better samples. One conjecture is that this allows the generator an opportunity to explore longer sequences in a free-running mode; in comparison, a random mask generally has shorter sequences of blanks to fill in, so the gain of GAN-training is not as substantial. We found that policy gradient methods were effective in conjunction with a learned critic, but the highly active research on training with discrete nodes may present even more stable training procedures.

We also found the use of attention was important for the in-filled words to be sufficiently conditioned on the input context. Without attention, the in-filling would fill in reasonable subsequences that became implausible in the context of the adjacent surrounding words. Given this, we suspect another promising avenue would be to consider GAN-training with attention-only models as in Vaswani et al. (2017).

In general we think the proposed contiguous in-filling task is a good approach to reduce mode collapse and help with training stability for textual GANs. We show that MaskGAN samples on a larger dataset (IMDB reviews) is significantly better than the corresponding tuned MaskMLE model as shown by human evaluation. We also show we can produce high-quality samples despite the MaskGAN model having much higher perplexity on the ground-truth test set.

ACKNOWLEDGEMENTS

We would like to thank George Tucker, Jascha Sohl-Dickstein, Jon Shlens, Ryan Sepassi, Jasmine Collins, Irwan Bello, Barret Zoph, Gabe Pereyra, Eric Jang and the Google Brain team, particularly the first year residents who humored us listening and commenting on almost every conceivable variation of this core idea.

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

## A    TRAINING DETAILS

Our model was trained with the Adam method for stochastic optimization (Kingma & Ba, 2015) with the default Tensorflow exponential decay rates of $\beta_1 = 0.99$ and $\beta_2 = 0.999$. Our model uses 2-layers of 650 unit LSTMs for both the generator and discriminator, 650 dimensional word embeddings, variational dropout. We used Bayesian hyperparameter tuning to tune the variational dropout rate and learning rates for the generator, discriminator and critic. We perform 3 gradient descent steps on the discriminator for every step on the generator and critic.

We share the embedding and softmax weights of the generator as proposed in Bengio et al. (2003); Press & Wolf (2017); Inan et al. (2017). Furthermore, to improve convergence speed, we share the embeddings of the generator and the discriminator. Additionally, as noted in our architectural section, our critic shares all of the discriminator parameters with the exception of the separate output head to estimate the value. Both our generator and discriminator use variational recurrent dropout (Gal & Ghahramani, 2016).

## B    ADDITIONAL SAMPLES

### B.1    THE PENN TREEBANK (PTB)

We present additional samples on PTB here.

#### B.1.1    CONDITIONAL SAMPLES

| Ground Truth | **the next day 's show <eos> interactive telephone technology has taken a new leap in <unk> and television programmers are** |
|---|---|
| MaskGAN | the next day 's show <eos> interactive telephone technology has taken a new leap in its retail business <eos> a |
| | the next day 's show <eos> interactive telephone technology has long dominated the <unk> of the nation 's largest economic |
| | the next day 's show <eos> interactive telephone technology has exercised a N N stake in the u.s. and france |
| MaskMLE | the next day 's show <eos> interactive telephone technology has taken a new leap in the complicate case of the |
| | the next day 's show <eos> interactive telephone technology has been <unk> in a number of clients ' estimates mountain-bike |
| | the next day 's show <eos> interactive telephone technology has instituted a week of <unk> by <unk> <unk> wis. auto |

We also consider filling-in on non-contiguous masks below.

| Ground Truth | **president of the united states ronald reagan delivered his <unk> address to the nation <eos> president reagan addressed several issues** |
|---|---|
| MaskGAN | president of the united states and congress delivered his <unk> address to the nation <eos> mr. reagan addressed several issues |
| MaskMLE | president of the united states have been delivered his <unk> address to the nation <eos> mr. reagan addressed several issues |

#### B.1.2    LANGUAGE MODEL (UNCONDITIONAL) SAMPLES

We present additional language model (unconditional) samples on PTB here. We modified SeqGAN to train and generate PTB samples using the same size architecture for the generator as in the MaskGAN generator and present samples here with MaskGAN samples.

| | |
|---|---|
| MaskGAN | a <unk> basis despite the huge after-tax interest income <unk> from $ N million <eos> in west germany N N |
| | the world 's most corrupt organizations act as a multibillion-dollar <unk> atmosphere or the metropolitan zone historic array with their |
| SeqGAN | are removed <eos> another takeover target lin 's directors attempted through october <unk> and british airways is allowed three funds |
| | cineplex odeon corp. shares made fresh out of the group purchase one part of a revised class of <unk> british |
| | there are <unk> <unk> and <unk> about the <unk> seed <eos> they use pcs are <unk> and their performance <eos> |

## B.2 IMDB MOVIE DATASET

We present additional samples on IMDB here.

### B.2.1 CONDITIONAL SAMPLES

| Ground Truth | **Pitch Black was a complete shock to me when I first saw it back in 2000 In the previous years I** |
|---|---|
| MaskGAN | Pitch Black was a complete shock to me when I first saw it back in 1979 I was really looking forward |
| | Pitch Black was a complete shock to me when I first saw it back in 1976 The promos were very well |
| | Pitch Black was a complete shock to me when I first saw it back in the days when I was a |
| MaskMLE | Black was a complete shock to me when I first saw it back in 1969 I live in New Zealand |
| | Pitch Black was a complete shock to me when I first saw it back in 1951 It was funny All Interiors |
| | Pitch Black was a complete shock to me when I first saw it back in the day and I was in |

### B.2.2 LANGUAGE MODEL (UNCONDITIONAL) SAMPLES

We present additional language model (unconditional) samples from MaskGAN on IMDB here.

**Positive**: Follow the Good Earth movie linked Vacation is a comedy that credited against the modern day era yarns which has helpful something to the modern day s best It is an interesting drama based on a story of the famed
**Negative**: I really can t understand what this movie falls like I was seeing it I m sorry to say that the only reason I watched it was because of the casting of the Emperor I was not expecting anything as
**Negative**: That s about so much time in time a film that persevered to become cast in a very good way I didn t realize that the book was made during the 70s The story was Manhattan the Allies were to

## C FAILURE MODES

Here we explore various failure modes of the MaskGAN model, which show up under certain bad hyperparameter settings.

## C.1  MODE COLLAPSE

As widely witnessed in GAN-training, we also find a common failure of mode collapse across various $n$-gram levels. The mode collapse may not be as extreme to collapse at a 1-gram level (dddddddd···) as described by Gulrajani et al. (2017), but it may manifest as grammatical, albeit, inanely repetitive phrases, for example,

> It is a very funny film that is very funny It s a very funny movie and it s charming It

Of course the discriminator may discern this as an out-of-distribution sample, however, in certain failure modes, we observed the generator to move between common modes frequently present in the text.

## C.2  MATCHING SYNTAX AT BOUNDARIES

We notice that the MaskGAN architecture often struggles to produce syntactically correct sequences when there is a hard boundary where it must end. This is also a relatively challenging task for humans, because the filled in text must not only be contextual but also match syntactically at the boundary between the blank and where the text is present over a fixed number of words.

> Cartoon is one of those films me when I first saw it back in 2000

As noted in this failure mode, the intersection between the filled in text and the present text is non grammatical.

## C.3  LOSS OF GLOBAL CONTEXT

Similar to failure modes present in GAN image generation, the produced samples often can lose global coherence, despite being sensible locally. We expect a larger capacity model can mitigate some of these issues.

> This movie is terrible The plot is ludicrous The title is not more interesting and original This is a great movie
> Lord of the Rings was a great movie John Travolta is brilliant

### C.4 $n$-GRAM METRICS MAY BE MISLEADING PROXIES

In the absence of a global scalar objective to optimize while training, we monitor various $n$-gram language statistics to assess performance. However, these only are crude proxies of the quality of the produced samples.

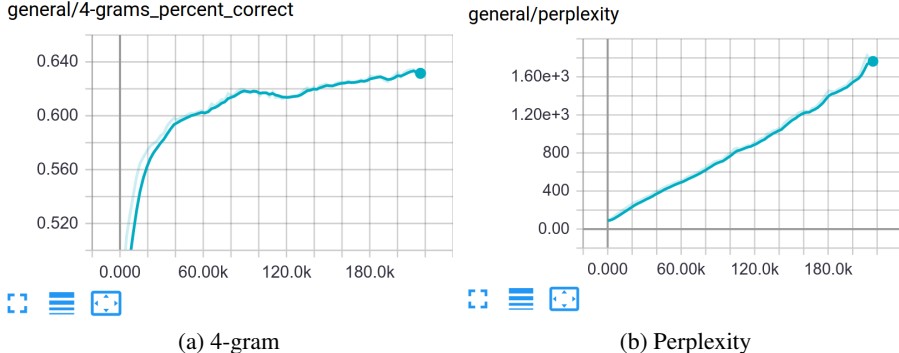

(a) 4-gram                    (b) Perplexity

Figure 2: Particular failure mode succeeding in the optimization of a 4-gram metric at the extreme expense of validation perplexity. The resulting samples are shown below.

For instance, MaskGAN models that led to improvements of a particular $n$-gram metric at the extreme expense of validation perplexity as seen in Figure 2 could devolve to a generator of very low sample diversity. Below, we produce several samples from this particular model which, despite the dramatically improved 4-gram metric, has lost diversity.

> It is a great movie It s just a tragic story of a man who has been working on a home
> It s a great film that has a great premise but it s not funny It s just a silly film
> It s not the best movie I have seen in the series The story is simple and very clever but it

Capturing the complexities of natural language with these metrics alone is clearly insufficient.

