# OpenReview forum: "MaskGAN: Better Text Generation via Filling in the _______"
_ICLR.cc/2018/Conference — Accept (Poster)_

### Official Review · AnonReviewer3 · 2017-11-27
**The paper discusses a mechanism of generating text samples using GAN and a in-filling mechanism of missing words conditional on the surrounding text. I feel the Reinforcement learning idea that has been introduced to employ a actor-critic training procedure could be computationally challenging and the motivation to do so is not explicitly clear.**

**Rating:** 7
**Confidence:** 3

**Review:**

Quality: The work focuses on a novel problem of generating text sample using GAN and a novel in-filling mechanism of words. Using GAN to generate samples in adversarial setup in texts has been limited due to the mode collapse and training instability issues. As a remedy to these problems an in-filling-task conditioning on the surrounding text has been proposed. But, the use of the rewards at every time step (RL mechanism) to employ the actor-critic training procedure could be challenging computationally challenging.

Clarity: The mechanism of generating the text samples using the proposed methodology has been described clearly. However the description of the reinforcement learning step could have been made a bit more clear.

Originality: The work indeed use a novel mechanism of in-filling via a conditioning approach to overcome the difficulties of GAN training in text settings. There has been some work using GAN to generate adversarial examples in textual context too to check the robustness of classifiers. How this current work compares with the existing such literature?

Significance: The research problem is indeed significant since the use of GAN in generating adversarial examples in image analysis has been more prevalent compared to text settings. Also, the proposed actor-critic training procedure via RL methodology is indeed significant from its application in natural language processing.

pros:
(a) Human evaluations applications to several datasets show the usefulness of MaskGen over the maximum likelihood trained model in generating more realistic text samples.
(b) Using a novel in-filling procedure to overcome the complexities in GAN training.
(c) generation of high quality samples even with higher perplexity on ground truth set.

cons:
(a) Use of rewards at every time step to the actor-critic training procure could be computationally expensive.
(b) How to overcome the situation where in-filling might introduce implausible text sequences with respect to the surrounding words?
(c) Depending on the Mask quality GAN can produce low quality samples. Any practical way of choosing the mask?

---

> ### Author Response · Authors · 2017-12-20
> **Re:  The paper discusses a mechanism [...]**
>
> Thank you for your review!
>
> *Importance and Computational Cost of Actor-Critic*
> We’d like to address your concern about the importance and the computational challenges of the actor-critic method.  We believe that this was a crucial component to get the results we did and it was achieved with no significant additional computational cost.
>
> In building architectures for this novel task, we were contending with both reinforcement learning challenges as well as GAN-mode collapse issues.  Specifically, variance in the gradients to the Generator was a major issue.  To remedy this, we simply added a value estimator as an additional head on the Discriminator.  The critic estimates the expected value of the current state, conditioned on everything produced before.  This is very lightweight in terms of additional parameters since we’re sharing almost all parameters with the Discriminator.  We found that using this reduced the advantage to the Generator by over an order of magnitude.  This was a critical piece of efficiently training our algorithm. We compared the performance of this actor-critic approach against a standard exponential moving average baseline and found there to be no significant difference in training step time.
>
> *Clarity*
> Thanks and we updated the writing to more clearly delineate the reinforcement learning training.
>
> *Originality*
> As far as we are aware, no work has considered this conditional task where a per-time-step reward is architected in.  Additionally, our use of an actor-critic methodology in GAN-training is a minimally explored avenue. Finally, the existing literature on textual adversarial examples focus on classifier accuracy and generally don't do human evaluations on the quality of the generated examples as we do.
>
> *Masking Strategy*
> We predominantly evaluated two masking strategies at training time.  One was a completely random mask and the other were contiguous masks, where blocks of adjacent words are masked.  Though we were able to train with both strategies, we found that the random mask was more difficult to train.  However, and more significantly, the random mask doesn’t share the primary benefit of GAN autoregressive text generation (termed free-running mode in the literature).  One can see this because for a given percentage of words to omit, a Generator given the random mask will fill-in shorter sequences autoregressively than the contiguous mask.  GAN-training allows our training and inference procedure to be the same, in contrast to teacher-forcing in the maximum likelihood training.  Therefore, we generally found it beneficial to allow the model to produce long sequences, conditioned on what it had produced before, rather than filling in short disjoint sequences or or even single tokens.

---

> > ### Author Response · Authors · 2017-12-30
> > **Additional baseline results**
> >
> > We additionally added results comparing MaskGAN and MaskMLE samples against those from a baseline LSTM language model. Tables 7 and 8 have been updated to include these results.

---

> > ### Comment · AnonReviewer3 · 2018-01-12
> > **Well revised and my concerns have been addressed**
> >
> > I am happy with the author's revision. The points I raised earlier have been addressed appropriately. The importance of the MaskGAN mechanism has been highlighted and the description of the reinforcement learning training part has been clarified.
> >
> > My other concern with the Masking strategy has been addressed and the two masking strategies have been described in detail.

---

### Official Review · AnonReviewer1 · 2017-11-28
**Very thorough empirical study**

**Rating:** 7
**Confidence:** 4

**Review:**

Generating high-quality sentences/paragraphs is an open research problem that is receiving a lot of attention. This text generation task is traditionally done using recurrent neural networks. This paper proposes to generate text using GANs. GANs are notorious for drawing images of high quality but they have a hard time dealing with text due to its discrete nature. This paper's approach is to use an actor-critic to train the generator of the GAN and use the usual maximum likelihood with SGD to train the discriminator. The whole network is trained on the "fill-in-the-blank" task using the sequence-to-sequence architecture for both the generator and the discriminator. At training time, the generator's encoder computes a context representation using the masked sequence. This context is conditioned upon to generate missing words. The discriminator is similar and conditions on the generator's output and the masked sequence to output the probability of a word in the generator's output being fake or real. With this approach, one can generate text at test time by setting all inputs to blanks.

Pros and positive remarks:
--I liked the idea behind this paper. I find it nice how they benefited from context (left context and right context) by solving a "fill-in-the-blank" task at training time and translating this into text generation at test time.
--The experiments were well carried through and very thorough.
--I second the decision of passing the masked sequence to the generator's encoder instead of the unmasked sequence. I first thought that performance would be better when the generator's encoder uses the unmasked sequence. Passing the masked sequence is the right thing to do to avoid the mismatch between training time and test time.

Cons and negative remarks:
--There is a lot of pre-training required for the proposed architecture. There is too much pre-training. I find this less elegant.
--There were some unanswered questions:
            (1) was pre-training done for the baseline as well?
            (2) how was the masking done? how did you decide on the words to mask? was this at random?
            (3) it was not made very clear whether the discriminator also conditions on the unmasked sequence. It needs to but
                  that was not explicit in the paper.
--Very minor: although it is similar to the generator, it would have been nice to see the architecture of the discriminator with example input and output as well.


Suggestion: for the IMDB dataset, it would be interesting to see if you generate better sentences by conditioning on the sentiment as well.

---

> ### Author Response · Authors · 2017-12-20
> **Re:  Very thorough empirical study**
>
> Thank you for your review!
>
> *Pretraining*
> We found evidence that this architecture could replicate simple data distributions without pretraining and found it could perform reasonably on larger data sets, however, in the interest of computational efficiency, we relied on pretraining procedures, similar to other work in this field. All our baselines also included pre-training.
>
> To test whether all the pretraining steps were necessary, we experimented with training MaskMLE and MaskGAN on PTB without initializing from a pretrained language model. The perplexity of the generated samples were 117 without pretraining and 126 with pretraining, showing that at least for PTB language model pretraining does not appear to be necessary.
>
> Models trained from scratch were found to more computationally intense.  By building off near state-of-the-art language models, we were able to rapidly iterate over architectures thanks to faster convergence.  Additionally, we were working at a word-level representation where our softmax is producing a distribution over O(10K)-tokens.  Attempting reinforcement learning methods from scratch on an ‘action space’ of this magnitude is prone to extreme variance.  The likelihood of producing a correct token and receiving a positive reward is exceedingly rare; therefore, the model spends a long time exploring the space with almost always negative rewards.  As a related and budding research avenue, one could consider the properties and characteristics of exclusively GAN-trained language models.
>
> *Masking Strategy*
> We predominantly evaluated two masking strategies at training time.  One was a completely random mask and the other was a contiguous mask, where blocks of adjacent words are masked.  Though we were able to train with both strategies, we found that the random mask was more difficult to train.  However, and more significantly, the random mask doesn’t share the primary benefit of GAN autoregressive text generation (termed free-running mode in the literature).  One can see this because for a given percentage of words to omit, a Generator given the random mask will fill-in shorter sequences autoregressively than the contiguous mask will.  GAN-training allows our training and inference procedure to be the same, in contrast to teacher-forcing in maximum likelihood training.  Therefore, we generally found it beneficial to allow the model to produce long sequences, conditioned on what it had produced before, rather than filling in short disjoint sequences or or even single tokens.

---

> > ### Author Response · Authors · 2017-12-30
> > **Additional baseline results**
> >
> > We additionally added results comparing MaskGAN and MaskMLE samples against those from a baseline LSTM language model. Tables 7 and 8 have been updated to include these results.

---

> > > ### Author Response · Authors · 2018-01-13
> > > **Re: Very thorough empirical study**
> > >
> > > Thanks again! Before the review process concludes, do you have any outstanding questions regarding our rebuttal which includes the additional experiments on pretraining and our chosen masking strategy?  In particular, we'd be interested in your opinion on the MaskGAN algorithm in light of evidence that it functions with less pretraining. Finally, our paper revision seeks to further strengthen our result by comparing against LSTM baselines.

---

> > > > ### Comment · AnonReviewer1 · 2018-01-13
> > > > **Re:  Very thorough empirical study**
> > > >
> > > > I acknowledge your rebuttal. I am updating my rating from 6 to 7 in light of it.

---

### Official Review · AnonReviewer2 · 2017-11-29
**Locally better, but globally not always better**

**Rating:** 7
**Confidence:** 5

**Review:**

This paper proposes MaskGAN, a GAN-based generative model of text based on
the idea of recovery from masked text.
For this purpose, authors employed a reinceforcement learning approach to
optize a prediction from masked text. Moreover, authors argue that the
quality of generated texts is not appropriately measured by perplexities,
thus using another criterion of a diversity of generated n-grams as well as
qualitative evaluations by examples and by humans.

While basically the approach seems plausible, the issue is that the result is
not compared to ordinary LSTM-based baselines. While it is better than a
conterpart of MLE (MaskedMLE), whether the result is qualitatively better than
ordinary LSTM is still in question.

In fact, this is already appearent both from the model architectures and the
generated examples: because the model aims to fill-in blanks from the text
around (up to that time), generated texts are generally locally valid but not
always valid globally. This issue is also pointed out by authors in Appendix
A.2.
While the idea of using mask is interesting and important, I think if this
idea could be implemented in another way, because it resembles Gibbs sampling
where each token is sampled from its sorrounding context, while its objective
is still global, sentence-wise. As argued in Section 1, the ability of
obtaining signals token-wise looks beneficial at first, but it will actually
break a global validity of syntax and other sentence-wise phenoma.

Based on the arguments above, I think this paper is valuable at least
conceptually, but doubt if it is actually usable in place of ordinary LSTM
(or RNN)-based generation.
More arguments are desirable for the advantage of this paper, i.e. quantitative
evaluation of diversity of generated text as opposed to LSTM-based methods.

*Based on the rebuttals and thorough experimental results, I modified the global rating.

---

> ### Author Response · Authors · 2017-12-20
> **Re:  Locally better, but globally not always better**
>
> Thank you for your review and comments!
>
> We reiterate your two primary concerns as the following:
> 1.  A standard LSTM-baseline of a non-masked task should be included.
> 2. The MaskGAN algorithm is enforcing only local consistency within text, but does not aid with global consistency.
>
> *Standard Baselines*
> To address your first concern, we added a thorough human evaluation of a language model (LM) LSTM baseline.  We use the samples produced from our Variational Dropout-LSTM language model and evaluate the resulting sample quality for both the PTB and IMDB datasets using Amazon Mechanical Turk.  You can see these results updated in our paper in Table 7 and Table 8.  We demonstrate that the MaskGAN training algorithm results in improvements over both the language model and the MaskMLE benchmarks on all three metrics: grammaticality, topicality and overall quality. In particular, MaskGAN samples are preferred over LM LSTM baseline samples, 58.0% vs 15.7% of the time for IMDB reviews.
>
> *Local vs. Global Consistency*
> In regards to your comment on Gibbs sampling, we do agree that this would likely be a valid and helpful technique for inference.  In our paper, we in-fill our samples autoregressively from left to right, as is conventional in language modeling.  (This approach allows for fast unconditional generation as with the LM baseline and is what our human evaluation is targeted at).  This autoregressive process relies on the attention module of our decoder in order to provide full context during the sampling process.  For instance, when the decoder is producing the probability distribution over token x_t, it is attending over the future context to create this distribution.  If the subject of the sentence is known to be a female leader and the model is generating a pronoun, the model has the ability to attend to the future context and select the correct gender-matched pronoun.  If the model fails to do this, a well-trained discriminator will ascribe a low reward to this pronoun selection which in turn will generate useful gradients through the attention mechanism.  We have observed this behaviour during preliminary experiments.  We argue that global consistency is built into this architecture but to solve the boundary problems in appendix C.2, allowing the autoregressive model decide when to stop instead of forcing it to output a fix number of words may resolve some of the syntactic issues.
>
> We also expand table 6 to show the diversity of the generated samples compared to a standard LM-LSTM.

---

### Decision · Program_Chairs · 2018-01-29
**ICLR 2018 Conference Acceptance Decision**

**Decision:**

Accept (Poster)

**Comment:**

This paper makes progress on the open problem of text generation with GANs, by a sensible combination of novel approaches.   The method was described clearly, and is somewhat original.   The only problem is the hand-engineering of the masking setup.